# Titanium Dioxide Nanoparticles Modulate Systemic Immune Response and Increase Levels of Reduced Glutathione in Mice after Seven-Week Inhalation

**DOI:** 10.3390/nano13040767

**Published:** 2023-02-18

**Authors:** Miroslava Lehotska Mikusova, Milena Busova, Jana Tulinska, Vlasta Masanova, Aurelia Liskova, Iveta Uhnakova, Maria Dusinska, Zora Krivosikova, Eva Rollerova, Radka Alacova, Ladislava Wsolova, Mira Horvathova, Michaela Szabova, Norbert Lukan, Zbynek Vecera, Pavel Coufalik, Kamil Krumal, Lukas Alexa, Vojtech Thon, Pavel Piler, Marcela Buchtova, Lucie Vrlikova, Pavel Moravec, Dusan Galanda, Pavel Mikuska

**Affiliations:** 1Faculty of Medicine, Slovak Medical University, 833 03 Bratislava, Slovakia; 2Institute of Hygiene and Epidemiology, First Faculty of Medicine, Charles University and General University Hospital in Prague, 121 08 Prague, Czech Republic; 3Health Effects Laboratory, Norwegian Institute for Air Research, 2007 Kjeller, Norway; 4Faculty of Public Health, Slovak Medical University, 833 03 Bratislava, Slovakia; 5Department of Environmental Analytical Chemistry, Institute of Analytical Chemistry of the Czech Academy of Sciences, 602 00 Brno, Czech Republic; 6RECETOX, Faculty of Science, Masaryk University, 625 00 Brno, Czech Republic; 7Laboratory of Molecular Morphogenesis, Institute of Animal Physiology and Genetics, Czech Academy of Sciences, 602 00 Brno, Czech Republic; 8Aerosol Chemistry and Physics Research Group, Institute of Chemical Process Fundamentals of the Czech Academy of Sciences, 165 00 Prague, Czech Republic; 9Public Health Authority of the Slovak Republic, Radiation Protection Department, 82645 Bratislava, Slovakia

**Keywords:** titanium dioxide nanoparticles, nanoparticle inhalation, immunotoxicity, immune response, lymphocytes, cytokines, inflammation, phagocytic activity and respiratory burst, antioxidant defense

## Abstract

Titanium dioxide nanoparticles (TiO_2_ NPs) are used in a wide range of applications. Although inhalation of NPs is one of the most important toxicologically relevant routes, experimental studies on potential harmful effects of TiO_2_ NPs using a whole-body inhalation chamber model are rare. In this study, the profile of lymphocyte markers, functional immunoassays, and antioxidant defense markers were analyzed to evaluate the potential adverse effects of seven-week inhalation exposure to two different concentrations of TiO_2_ NPs (0.00167 and 0.1308 mg TiO_2_/m^3^) in mice. A dose-dependent effect of TiO_2_ NPs on innate immunity was evident in the form of stimulated phagocytic activity of monocytes in low-dose mice and suppressed secretory function of monocytes (IL-18) in high-dose animals. The effect of TiO_2_ NPs on adaptive immunity, manifested in the spleen by a decrease in the percentage of T-cells, a reduction in T-helper cells, and a dose-dependent decrease in lymphocyte cytokine production, may indicate immunosuppression in exposed mice. The dose-dependent increase in GSH concentration and GSH/GSSG ratio in whole blood demonstrated stimulated antioxidant defense against oxidative stress induced by TiO_2_ NP exposure.

## 1. Introduction

Titanium dioxide nanoparticles (TiO_2_ NPs) are one of the most widely produced nanomaterials in the world. They are used in a wide range of applications, including coatings, paints, plastics, paper, inks, textiles, cosmetics, pharmaceuticals, food, agricultural production, environmental remediation, wastewater treatment, antibacterial agents, electronics, catalysts, solar cells, biomedical applications, etc. [1,2]. The mass production and use of these nanomaterials has great potential for human exposure and release into the environment [3]. 

Inhalation is the primary route of occupational exposure to TiO_2_ NPs. Inhalation exposure is also possible with the use of antimicrobial sprays containing TiO_2_ NPs [4]. The residence time of NPs in air may differ from that of larger (fine) particles of the same material. In general, the smaller and lighter a particle is, the longer it remains in the atmosphere [5].

Compared to their fine particles, TiO_2_ NPs have different physicochemical properties that may alter their bioactivity. TiO_2_ particles with a smaller diameter cause more significant pulmonary inflammation at the same mass burden. However, after equalization based on particle surface area, the lung response to nanoscale TiO_2_ is similar to that of fine TiO_2_ particles [4]. Most studies on the effects of TiO_2_ NPs on the pulmonary system have shown that TiO_2_ NPs can cause histopathological alterations, inflammation, immune system dysfunction, genotoxicity, and carcinogenesis [6]. In addition to the lungs, the function of many other tissues such as the brain, liver, kidney, spleen, and blood may also be affected after inhalation exposure [4,6,7,8].

The US National Institute for Occupational Safety and Health (NIOSH) recommends an airborne exposure limit (REL) of 0.3 mg/m^3^ for TiO_2_ NPs, which is eight times lower than the REL for fine TiO_2_ [9]. The real data from different factories producing TiO_2_ NPs show occupational exposure to TiO_2_ in the range of 0.10 to 5.99 mg/m^3^ [9,10] with the percentage of TiO_2_ particles with diameter less than 100 nm up to 42.4% [9]. In 2006, the International Agency for Research on Cancer (IARC) classified TiO_2_ as a possible inhalation carcinogen (Group 2B) for humans [11]. Since 2020, the European Union has also classified TiO_2_ as a suspected carcinogen by inhalation. This classification applies only to mixtures in powder form containing 1% or more TiO_2_ particles with an aerodynamic diameter ≤ 10 µm [12]. TiO_2_ is currently approved as a UV filter in cosmetic products, including nanomaterial, at a maximum concentration of 25% when applied to healthy, intact or sunburned skin. In 2022, the European Commission withdrew the EU approval for the use of TiO_2_ in food as colorant E171 [13].

TiO_2_ NPs are able to pass through biological membranes, enter the cells, and accumulate in tissues and organs, where they can exert toxic effects [14]. Although the integral translocation fraction of TiO_2_ NPs across the air–blood barrier after inhalation is limited (0.15), it has been shown that inhaled TiO_2_ NPs can be translocated via the lung–red blood cell–target organ axis, metabolized by the liver, and excreted via the urinary tract [7]. Although dissolution of NPs was found to be negligible, translocated TiO_2_ NPs are predominantly excreted in the urine as disagglomerated primary particles and/or smaller agglomerates [15]. The limited elimination of TiO_2_ NPs from the internal organs, related to their low solubility, may lead to their accumulation in tissues upon repeated exposure [16,17]. TiO_2_ NPs can affect the integrity of the blood–brain barrier (BBB) due to their persistence in endothelial cells or via infiltration of immune cells, resulting in breakdown of BBB [18]. NPs can also enter the brain via the olfactory route when administrated by inhalation or intranasally. Once in the brain, they may cause impairment of neurons and glial cells and lead to CNS dysfunction [19,20,21]. 

TiO_2_ NPs primarily cause adverse biological effects by inducing oxidative stress associated with increased production of reactive oxygen species (ROS), oxidation products, and depletion of cellular antioxidants [22]. NPs can generate ROS directly from their surface or by activating macrophages, leading to increased inflammation. They can also alter mitochondrial function, stimulate cytokine production, and cause DNA alternations, cellular damage, and disease [23]. Free radicals induced by NPs alter glutathione (GSH) to its oxidized form GSSG. GSH is known to be a nonenzymatic antioxidant and a free radical scavenger. It maintains the cellular redox state and protects cells from oxidative damage [24]. Decreased GSH levels and increased lipid peroxidation and ROS levels after exposure to TiO_2_ NPs may lead to cell death. High concentrations of TiO_2_ NPs causing high oxidative stress may lead to cell damage responses, whereas low levels of oxidative stress may cause inflammation, which might be stimulated by activation of ROS signaling pathways [25]. The results of a recent study support the theory that TiO_2_ NPs induce the formation of ROS and inhibit cellular enzymatic mechanisms, including effects on GSH levels [26]. 

Numerous in vitro and in vivo studies have demonstrated the potential of TiO_2_ NPs for immunomodulatory effects [27,28,29,30,31,32]. However, only a few have studied the effects of inhalation exposure on the systemic immune response [8,33,34]. Inhaled TiO_2_ NPs have been found to affect the number of immune cells in the blood, such as monocytes, granulocytes, lymphocytes, and platelets [7,33]. They are readily taken up by cells of the immune system, can accumulate in peripheral lymphoid organs such as the spleen and lymph nodes, and influence various manifestations of immune cell activity, including cytokine production [18,34,35] and proliferation [8,35].

Several methods have been proposed to generate TiO_2_ nanoparticles (NPs). In general, TiO_2_ can be synthetized either by the thermal hydrolysis [36,37] or by the oxidation of TiCl_4_ [36,38,39,40], or by the thermal oxidation [41,42,43] or pyrolysis [40,43,44,45] of titanium tetraisopropoxide (TTIP). The generation TiO_2_ by TTIP oxidation is accompanied only by production of CO_2_ and H_2_O, unlike side toxic products such as HCl or Cl_2_ in the case of TiCl_4_ conversion, or C_3_H_6_ during the pyrolysis of TTIP.

In this study, we investigated the potential effects of TiO_2_ NPs on immune response and antioxidant defense. Here, we report the effects of subchronic inhalation exposure to two different concentrations of these NPs (0.00167 and 0.1308 mg TiO_2_/m^3^) carried out in whole-body inhalation chambers continuously for seven weeks. Although several inhalation studies on the systemic immune effects of TiO_2_-NPs have been conducted previously, they were performed by intratracheal [8,33,35] or intranasal [34] instillation. Compared with instillation, inhalation better mimics the physiological pathway of inhalation exposure to nanomaterials. 

Because reliable data from studies with doses relevant to human exposure are still lacking, the concentrations of TiO_2_ NPs used in the present study were lower than in previous studies, i.e., closer to a realistic exposure scenario in occupational settings [9,10].

The aim of our study was to investigate the effects of TiO_2_ NPs on the systemic immune response and antioxidant defense in the blood of mice (GSH, GSSG, and GSH/GSSG ratio) after seven weeks of continuous inhalation.

## 2. Materials and Methods

### 2.1. Animals

Adult female mice (ICR line, six weeks old, average body weight 24 g) were obtained from Masaryk University (Brno, Czech Republic). Prior to the experiment, the animals were acclimated to laboratory conditions for one week. A commercial diet and water were provided ad libitum. All animal experiments were performed in accordance with the Guidelines for the Care and Use of Laboratory Animals of the Institute of Analytical Chemistry of the Czech Academy of Sciences (Ministry of Agriculture of the Czech Republic, No. 10031/2013-MZE-17214) and approved by the Animal Ethics Committee of the Institute of Animal Physiology and Genetics of the Czech Academy of Sciences (No. 081/2010). 

### 2.2. Preparation of NPs

TiO_2_ NPs were continuously generated in situ via the aerosol route in a hot-wall tubular flow reactor by thermal decomposition of titanium tetraisopropoxide (TTIP) in a vertically oriented furnace (Carbolite TZF 15/50/610) at a temperature of 751 °C [46]. The vapors of TTIP were generated from the liquid form of TTIP in a saturator at 24 °C and the released vapors were transported into the flow reactor with a nitrogen stream (purity 99.9995%; flow rate 0.85 L/min). Before entering the reactor, it was diluted with another nitrogen stream (purity 99.9995%; flow rate 0.90 L/min). In parallel, a stream of oxygen (99.9996%; flow rate 0.40 L/min) was introduced into the reactor to oxidize the organic part of the TTIP. At the outlet of the reactor, the TiO_2_ NPs transported in a mixture of nitrogen and oxygen (flow rate 2.15 L/min) were mixed with air (flow rate 5.00 L/min). The generated TiO_2_ NPs were consecutively diluted with U-HEPA-filtered air (flow rate 20.0 L/min) and used for whole-body inhalation experiments in two inhalation cages with different concentrations of TiO_2_ NPs. All flow rates were regulated with Aalborg GFCS electronic mass flow controllers. 

The size and shape of the generated TiO_2_ NPs were characterized by transmission electron microscopy (TEM, Magellan 400 L XHR microscope, FEI Company, Hillsboro, OR, USA). TiO_2_ NPs were collected by electrostatic precipitation by using a nanometer aerosol sampler (Model 3089, TSI, Shoreview, MN, USA) on TEM grids (copper S160-4, 3 mm diameter, 400 mesh grid, Agar Scientific, Electron Technology, Stansted, Essex, UK). Samples were analyzed by scanning transmission electron microscopy in STEM mode. The micrograph (Figure 1) showed that the TiO_2_ NPs in the air inside the inhalation cage, measured by a Scanning Mobility Particle Sizer (model 3972L, TSI, USA), consisted mainly of agglomerates of primary particles with diameters ranging from 2 to 6 nm and small number of larger particles with size up to a diameter of 15 nm was also found. 

The surface areas of the generated TiO_2_ NPs were 1.94 × 10^8^ and 1.16 × 10^10^ nm^2^/cm^3^. The surface area was calculated from the NP size distribution data by using SMPS software [47]. 

### 2.3. Exposure to TiO_2_ NPs

Animals were exposed to TiO_2_ NPs in a special inhalation chamber [48], which was made of glass and stainless steel and contained four stainless steel inhalation cages. A control group of animals was housed in the inhalation chamber in a cage without exposure to NPs. 

The air conditioning system maintained constant parameters of the air flowing through the inhalation cages (i.e., temperature, relative humidity, and pressure). Air parameters were measured and recorded online at one-minute intervals. Lighting was set to 12 h of light and 12 h of darkness. The behavior and health status of the mice were continuously monitored with a camera system. The distributions of generated NPs were measured directly in the inhalation cages with a scanning mobility particle sizer. The size distribution is shown in Figure 2 for low and high concentrations of TiO_2_ NPs, respectively. The number concentrations of TiO_2_ NPs were 5.06 × 10^4^ (mode 32.2 nm, geometric mean diameter 29.6 nm, geometric standard deviation 1.64) and 1.51 × 10^6^ particles/cm^3^ (mode 37.2 nm, geometric mean diameter 30.3 nm, geometric standard deviation 1.85). The average mass concentrations of TiO_2_ NPs were 1.67 and 130.8 µg TiO_2_/m^3^ for low and high number concentrations, respectively. The special feeding device (a tube closed at the top from which the feed falls down into the feeder) was designed to minimize oral ingestion of NPs resulting from contamination of commercial feed granules by adsorption of TiO_2_ NPs on their surface. 

Mice were exposed to TiO_2_ NPs continuously for seven weeks, 24 h/day, seven days/week. Control animals were exposed to the same air as the experimental groups, only without the NP supplement. The estimated total deposited dose during the seven-week inhalation period was 0.012 and 0.958 µg TiO_2_ per gram of body weight of mice [49]. At the end of the inhalation experiment, the mice were directly decapitated and immediately dissected. Blood and organs were isolated and subjected to immune system and antioxidant defense examination. 

### 2.4. Immunoassays

#### 2.4.1. Phagocytic Activity of Granulocytes and Monocytes and Respiratory Burst of Phagocytes

The assay was performed as described elsewhere [50]. Heparinized whole blood from mice was mixed with hydroethidine and incubated at 37 °C for 15 min. Subsequently, samples were incubated with fluorescein-labeled *Staphylococcus aureus* (SA, Invitrogen, Waltham, MA, USA) for an additional 15 min at 37 °C. Then they were placed on ice and cold lysis solution was added. For the control tubes, the SA bacteria were added after the lysis solution. EPICS XL flow cytometer (Beckman Coulter, Brea, CA, USA) was used to analyze the samples by using forward and side scatter gates. The percentage of phagocytic monocytes, phagocytic granulocytes, and granulocytes with respiratory burst was measured in duplicate. The results were analyzed by flow cytometry as follows: % of phagocytic granulocytes = phagocytic granulocytes/all granulocytes.

#### 2.4.2. Phenotypic Analysis of Spleen, Thymus, and Bone Marrow

The spleen, thymus, and bone marrow were placed in complete RPMI 1640 culture medium (Sigma-Aldrich, St. Louis, MO, USA), as previously described by Tulinska et al. [50]. The following antibodies, purchased from eBioscience (San Diego, CA, USA), were used to stain the cells: Anti-Mouse CD3e PE, Anti-Mouse CD4 FITC, Anti-Mouse CD8a PerCP-eFluor^®^ 710, Anti-Mouse CD19 FITC and Anti-Mouse CD335 PerCP-eFluor^®^ 710, isotypic controls (IK): Armenian Hamster IgG IK PE, Rat IgG2aƙ IK FITC, Rat IgG2aƙ IK PerCP-eFluor^®^ 710. The concentrations recommended by the manufacturer were applied. Samples were analyzed by using the EPICS XL flow cytometer (Beckman Coulter, St. Louis, MO, USA). The percentage of CD3e^+^ in spleen, thymus, and bone marrow, of CD3e^+^CD4^+^ and CD3e^+^CD8a^+^ in spleen and thymus, and of CD3^−^CD19^+^ and CD3e^−^CD335^+^ in spleen and bone marrow were analyzed in duplicate.

#### 2.4.3. In Vitro Production of Cytokines and Chemokines

Spleen cells were cultured with the mitogen concanavalin A (Con A) (Sigma-Aldrich, St. Louis, MO, USA) in microtitrate plates at 37 °C for 48 h. The supernatants were stored at −70 °C. The ProcartaPlex^®^ Mouse cytokine and chemokine panel (eBioscience, San Diego, CA, USA) was used to measure the levels of Eotaxin/CCL11, granulocyte-macrophage colony-stimulating factor (GM-CSF), chemokine growth-regulated protein alpha (GRO-α/KC/CXCL1), interferon-γ (IFN-γ), interleukin (IL)-1β, IL-2, IL-4, IL-5, IL-6, IL-9, IL-10, IL-12p70, IL-13, IL-17A, IL-18, IL-22, IL-23, and IL-27, interferon gamma-inducible protein 10 (IP-10/CXCL10), monocyte chemotactic protein-1 (MCP-1/CCL-2), MCP-3/CCL7, macrophage inflammatory protein-1α (MIP-1α/CCL3), MIP-1β/CCL4, MIP-2/CCL8, RANTES/CCL5, and tumor necrosis factor-α (TNF-α) according to the manufacturer’s instructions by using the Luminex xMAP instrument (Luminex-Corporate, Austin, TX, USA). Values extrapolated beyond the calibration curve (cc) and values outside the cc range (below, above) were discarded. Levels of GRO-α/KC/CXCL1, IL-1β, IL-5, IL-9, IL-12p70, IL-22, IL-23, IL-27 were below the limit of detection and are therefore not presented.

### 2.5. Antioxidant Status, Reduced Glutathione, and Oxidized Glutathione

Concentrations of reduced (GSH) and oxidized (GSSG) glutathione as markers of oxidative stress and antioxidant defense were determined by using the Bioxytech^®^ GSH/GSSG-412™ assay kit (Oxis International, Inc., Portland OR, USA) according to the method of Ellman [51], modified by Tietze [52]. Immediately after blood collection into the EDTA-containing plastic tubes, 100 µL of each whole blood sample for GSSG determination was transferred to a 1.5-mL microtube (Eppendorf, Hamburg, Germany) containing 10 µL of the scavenger M2VP (1-methyl-2-vinyl-pyridium-trifluoromethane sulfonate) to prevent oxidation of GSH to GSSG during sample preparation. Next, 50 µL of the whole blood sample for GSH determination was transferred into a 1.5-mL microtube (Eppendorf) without treatment. All samples were frozen at −80 °C until analysis. Before analysis, blood samples were thawed and mixed. After all procedures and reaction with Ellman’s reagent (5,5′-dithiobis-2-nitrobenzoic acid, DTNB), samples were measured in duplicate by using a spectrophotometric reader at 412 nm (Epoch, BioTek, Santa Clara, CA, USA). The antioxidant status of the animals was assessed by blood GSH and GSSG concentrations and GSH/GSSG ratio. The concentrations of GSH and GSSG are expressed in µmol/L. The GSH/GSSG ratio was calculated according to the instructions of the assay kit.

### 2.6. Statistical Analysis

SPSS software was used for statistical analysis. Data are presented as mean ± standard error of the mean (SEM). Multiple measurements from each individual were averaged and used as a single value for analysis. The Shapiro–Wilk test was applied to test the normality of the data distribution. Oxidative stress data (GSH, GSSG) were tested to identify and eliminate outliers (Grubbs’ test). A Student *t*-test (for normally distributed data sets) or Mann–Whitney tests (for nonnormally distributed datasets) were used to determine differences between exposed and control groups. Differences with *p* < 0.05 were considered statistically significant. *P* values are given as follows: * *p* < 0.05; ** *p* < 0.01; *** *p* < 0.001.

## 3. Results

### 3.1. Phagocytic Activity of Blood Monocytes, Granulocytes, and Respiratory Burst

Phagocytic activity of cells was evaluated by flow cytometry. TiO_2_ NP exposure significantly stimulated the phagocytic activity of monocytes in mice exposed to the low dose compared with controls (Figure 3). No significant differences in phagocytic activity of granulocytes and respiratory burst of granulocytes were recorded. 

### 3.2. Phenotypic Analysis of Spleen, Thymus, and Bone Marrow

Phenotypic analysis of cells in the spleen, thymus, and bone marrow was performed by flow cytometry. The results of the measurement of CD3^+^ (T-lymphocytes), CD3^+^CD4^+^ (T-helper lymphocytes), CD3^+^CD8^+^ (T-cytotoxic lymphocytes), CD3^−^CD19^+^ (B-lymphocytes), and CD3^−^CD335^+^ (natural killer (NK) cells) are summarized in Table 1. The effect of seven-week inhalation of TiO_2_ NPs was manifested by a significantly decreased percentage of T-lymphocytes in the spleen in both dose groups vs. controls. In mice exposed to the low dose of TiO_2_ NPs, the percentage of T-helper lymphocytes in the spleen was also reduced compared with controls. The percentages of splenic CD3^+^CD8^+^, CD3^−^CD19^+^, and CD3^−^CD335^+^ cells in exposed mice were not different from those of controls. Phenotypic analysis of the thymus and bone marrow showed no significant difference between the exposed and control mice or between the low-dose and high-dose groups for any of the parameters examined. 

### 3.3. In Vitro Production of Cytokines

The in vitro production of several key cytokines was examined by the luminescence method. Cytokine levels measured in cultures from spleen cells of mice are shown in Table 2. A seven-week inhalation of TiO_2_ NPs in mice resulted in a significant decrease in the levels of IL-4 and IL-18 in the high-dose group compared with controls. Moderately reduced levels of IL-2, IL-10, IL-17A, IFN-γ, IL-6, TNF-α, GM-CSF, IL-13, and the chemokines MIP-1α, MIP-1β, MIP-2, and RANTES were found in mice receiving a high dose. In mice receiving a low dose of NPs, a marked but nonsignificant increase in the secretion of monocyte chemoattractant proteins (MCPs) MCP-1 and MCP-3 was found. No differences in the levels of IL-12p70, eotaxin, and IP-10 were observed.

### 3.4. Antioxidant Status of Blood—Reduced Glutathione and Oxidized Glutathione

The glutathione system is one of the defense mechanisms against the effects of xenobiotics and free radicals. The overall status of antioxidant protection of the organism was evaluated by GSH and GSSG content, and GSH/GSSG ratio in blood samples. The experimental group exposed to the low concentration of TiO_2_ NPs showed a 32% increase in GSH concentration compared with the control group (1 037.6 ± 469.2 vs. 785.5 ± 312.3 µmol/L), but the difference was not significant. In the high-dose animals, a 74% increase in GSH concentration (1 368.6 ± 306.8 µM) was statistically significant (*p* = 0.006) (see Table 3). On the other hand, no significant effect on GSSG content was observed in the low-dose and high-dose experimental groups compared with the control group. Regarding the GSH/GSSG ratio, a significant dose-dependent increase was observed. 

## 4. Discussion

Our study extends the number of inhalation studies on TiO_2_ NPs by using a whole-body inhalation chamber model, which is otherwise extremely rare (see Appendix A) [7,53]. In addition, none of the previous studies addressed the effects on the immune response in exposed animals. In this study, the profile of lymphocyte markers, functional immunoassays, and markers of antioxidant defense were selected to evaluate the potential adverse effects of inhalation of TiO_2_ NPs in mice exposed for seven weeks. Many of the immunotoxic effects of engineered nanomaterials are mediated by direct interaction with the innate immune system [54]. Macrophages play a critical role in immune surveillance of pathogens and clearance of inhaled particles and fibers [55]. In our study, a seven-week inhalation of TiO_2_ NPs stimulated the phagocytic activity of monocytes and substantially, but not significantly, enhanced the secretion of monocyte chemoattractant proteins MCP-1 (200%) and MCP-3 (167%) in low-dose mice. Activation of phagocytic activity of monocytes suggests significant action to clean NPs from the organism. Active phagocytosis of the TiO_2_ NPs was confirmed by cytoplasmic proteome analysis in macrophages derived from human monocytes [56]. 

In addition to interfering with phagocytosis, damage to the pulmonary macrophages may be manifested by decreased chemotactic ability and MHC-class II expression on the cell surface, as well as increased secretion of nitric oxide and upregulation of inflammatory proteins (MIP and MCPs) after intratracheal instillation of TiO_2_ NPs [18,35,57].

Oxidative stress and cytotoxicity seem to be the underlying mechanisms of the effect of TiO_2_ NPs on macrophages. The observed oxidative stress was associated with the formation of intracellular ROS [56,58]. Cytotoxicity resulted in changes in the molecular patterns of proteins [56,58] and nucleic acids [58], remodeling of the cytoskeleton [56,59], damage to organelles [58], and formation of large vacuoles [59]. 

The assessment of the acquired immune response in our study included subpopulations of lymphocytes and NK cells in spleen, thymus, and bone marrow. The analysis showed a significant reduction in the percentage of T-cells in the splenic lymphocytes in both TiO_2_ NPs exposed groups. T-helper cells were significantly decreased only in the low-dose mice. The depletion of T-cells suggests that exposure to TiO_2_ NPs may lead to immunosuppression, as the immune system fails to respond with an increase in specific T-cell populations [60,61]. 

The complex functions of immune cells present in the spleen were studied by in vitro production of cytokines and chemokines. The suppressed percentage of T-lymphocytes and T-helper cells directed our attention to the production of IL-2, IL-10, and IL-17A. Indeed, the levels of all three cytokines were moderately decreased in the high-dose group of mice (63%, 60%, 48%). The decrease in IL-17A was close to statistical significance (*p* = 0.063). Moreover, the reduced IL-4 together with the moderately decreased lymphocyte-driven cytokines IFN-γ (40%), GM-CSF (39%), IL-13 (44%), and chemokine RANTES (63%, *p* = 0.056) observed in mice exposed to the high dose of TiO_2_ NPs is a clear indication of the toxicity of TiO_2_ NPs. The overall picture is complemented by the suppressed secretory function of macrophages, as evidenced by significantly lower levels of the proinflammatory cytokine IL-18, accompanied by moderately decreased levels of TNF-α (57%), IL-6 (46%), MIP-1α (60%), MIP-1β (52%), and MIP-2 (70%) in the splenic supernatants of the high-dose animals. Similar to our results, published data on cytokine production induced by TiO_2_ NPs in a long-term inhalation study in rats showed a decrease in blood IFN-γ and TNF-α levels [62]; however, no changes [8] or stimulation [34,35,63] of cytokine expression/release were observed.

It is evident that TiO_2_ NPs can modulate the immune response through multiple pathways with different results [30,64,65,66]. The suppressive effects of TiO_2_ nanostructure materials may be mediated by impairment of lymphocyte development [30] and proliferation [30,66] with suppression of Th1-cytokines [66] or decrease in inflammatory-related gene transcription [63]. Finally, the suppressive effects of TiO_2_ nanomaterials in the context of the immune response are best demonstrated by using animal models of complex host resistance to infection and tumors. Exposure to TiO_2_ NPs exacerbated pneumonia in mice infected with respiratory syncytial virus [67] and significantly increased tumor growth in mice implanted with B16F10 melanoma cells [30]. 

We also examined the overall antioxidant status of the organisms and showed a dose-dependent increase in GSH concentration in the whole blood of mice exposed to TiO_2_ NPs. The significantly increased GSH levels may indicate the self-regulation of some enzymes and antioxidants as a defense response to oxidative stress stimulated by exposure to TiO_2_ NP. Glutathione is one of the most important antioxidants abundantly present in cells and biological fluids throughout the body. The reduced form of glutathione reacts with ROS to form GSSG, which is rapidly regenerated. 

Similar to our results, Wang et al. [68] reported an increase in GSH levels in the rat synovium after intraarticular injection of TiO_2_ NPs. They also observed upregulated GSH-peroxidase, superoxide dismutase, and lipid peroxidation. In a short-term study, Sangeetha et al. [69] found increased GSH content and lipid peroxidation in the liver and spleen of mice treated orally with 1.6 mg/kg _b.w._ of TiO_2_ NPs. Some in vitro studies have also shown elevated GSH levels after TiO_2_ NP exposure [70,71]. The effect of TiO_2_ NP exposure in human bronchial epithelial cells (increase or decrease in GSH level) depended on the dose, size, and agglomeration of TiO_2_ NPs [71].

On the other hand, our results are not consistent with the results of other subchronic inhalation studies, in which a decrease or no change in GSH levels in blood or tissues was observed. However, the doses administered in these experiments were much higher compared to our study. Liu et al. [62] found decreased GSH levels in the blood of rats after intratracheal instillation of TiO_2_ NPs (3.5 and 17.5 mg/kg _b.w._), whereas Relier et al. [72] reported no change in total glutathione levels in the plasma, lung, and liver of rats after intratracheal exposure to TiO_2_ NPs (0.5, 2.5 and 10 mg/kg _b.w._). Wang et al. [73,74] also observed no change in GSH levels in the brain of female mice intranasally administered 500 µg of a TiO_2_ NP suspension every other day for 20 or 30 days, whereas GSH levels were significantly increased 10 days after exposure. 

We suppose that the increase in GSH content observed in the present study may be a response to the excessive formation of ROS, as defense mechanisms were activated due to TiO_2_ NP exposure and GSSG reduction was enzymatically increased in favor of GSH. Considering the low dose of TiO_2_ NPs used in our study, we suppose that the increased GSH levels were likely the primary means by which cells prevented lipid hydroperoxide formation. The enhancement of enzymatic reactions and the increase in GSH production could be a triggering mechanism to support defense against the deleterious effects of TiO_2_ NPs. This finding is in agreement with the results of Carmo et al. [75] who observed increased GSH content and no change in antioxidant enzyme activity in fish liver after acute exposure (48 h) to 1, 5, 10, and 50 mg/L TiO_2_ NPs, whereas subchronic exposure (14 days) to the same TiO_2_ NP concentrations decreased superoxide dismutase activity and increased glutathione-S-transferase activity and GSH content. Nevertheless, the animal strain and genetic background used, the dose and duration of exposure, and the size and agglomeration of TiO_2_ NPs may affect the antioxidant defense of the organism.

We exposed animals to TiO_2_ NPs in whole-body inhalation chambers to simulate natural conditions. In this regard, it should be mentioned that in this procedure TiO_2_ NPs may adhere to the walls of the polycarbonate boxes in the inhalation cages or come into contact with the food or various body parts of the mice such as the respiratory tract, the olfactory system, the fur or skin of the mouse, and also the eyes. As a result, mixed inhalation and oral uptake of nanoparticles is possible. To minimize the oral intake of TiO_2_ NPs, the feed was placed in a special feeding device that we designed to protect the granules from nanoparticle contamination. However, mutual licking of the fur by the animals cannot be avoided. Licking of nanoparticles adhering to the walls of the boxes was minimized by cleaning the walls during regular feed replenishment and bedding changes.

## 5. Conclusions

TiO_2_ NPs are used in large quantities in various industries worldwide, and therefore experimental studies on possible toxic effects are important to ensure human safety. Our results showed a dose-dependent effect of TiO_2_ NPs by inhalation on innate immunity, which was manifested by stimulated phagocytic activity of monocytes in low-dose mice and suppressed secretory function of monocytes in high-dose animals. The effect of TiO_2_ NPs on adaptive immunity, which was manifested in the spleen by a decrease in the percentage of T-cells, a reduction in T-helper cells, and a dose-dependent decrease in cytokine production by lymphocytes, may indicate immunosuppression. The dose-dependent increase in GSH concentration and GSH/GSSG ratio suggests self-regulation of some enzymes and antioxidants as a defence response to oxidative stress stimulated by TiO_2_ NP exposure.

In summary, our results show that relatively low doses of TiO_2_ NPs have significant immunomodulatory effects in mice affecting innate and adaptive immune responses. This indicates the potential risk of adverse health effects from inhalation of TiO_2_ NPs. This information may be useful for risk assessment of exposure to TiO_2_ NPs.

## Figures and Tables

**Figure 1 nanomaterials-13-00767-f001:**
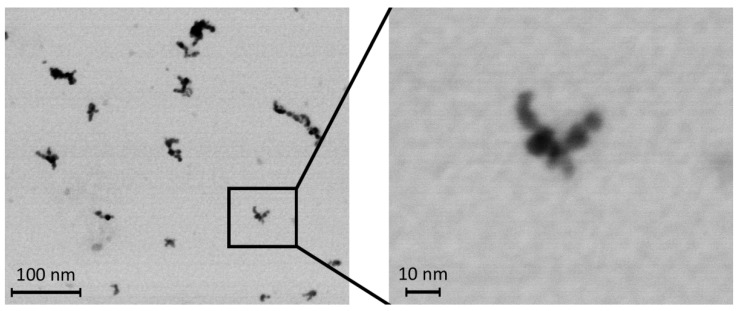
Transmission electron micrograph of TiO_2_ NPs deposited on the TEM grid.

**Figure 2 nanomaterials-13-00767-f002:**
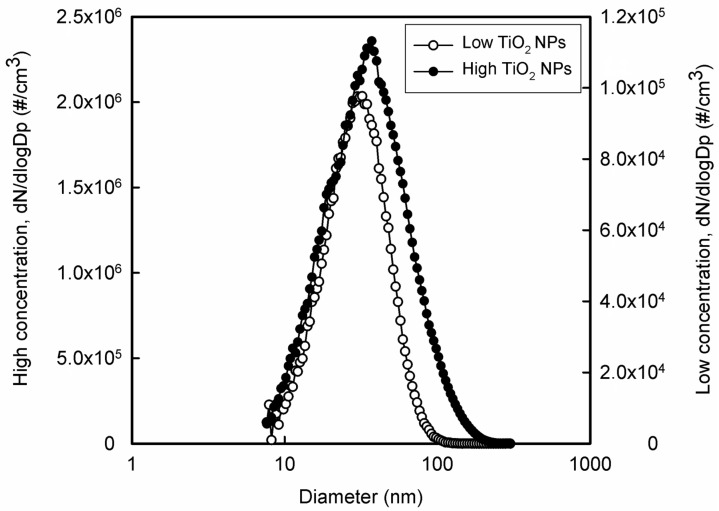
The number size distribution of TiO_2_ NPs for low NP concentration (5.06 × 10^4^ particles/cm^3^) and high NP concentration (1.51 × 10^6^ particles/cm^3^). x-axis, particle diameter (a logarithmic scale); y-axis, number concentration of particles (the number concentration is normalized by the size range of particles).

**Figure 3 nanomaterials-13-00767-f003:**
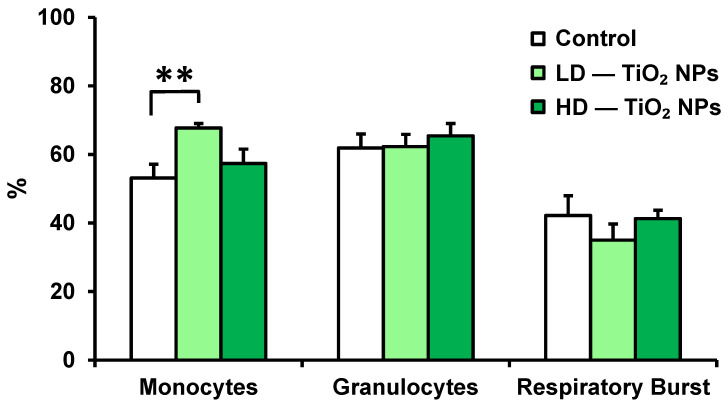
Phagocytic activity and respiratory burst of leukocytes. Phagocytic activity of monocytes and granulocytes in the blood of mice exposed to TiO_2_ NPs for seven weeks was evaluated by using ingestion of fluorescein-labeled *Staphylococcus aureus*, and the respiratory burst was monitored by using hydroethidine by flow cytometry. Control, control group (n = 10); LD–TiO_2_ NPs, group exposed to the low dose of TiO_2_ NPs (0.012 µg TiO_2_/g _b.w._) (n = 7); HD–TiO_2_ NPs, group exposed to the high dose of TiO_2_ NPs (0.958 µg TiO_2_/g _b.w._) (n = 10). Results are expressed as the percentage of phagocytic activity and respiratory burst. Bars indicate mean group activity of blood cells (mean + SEM). Significance: ** *p* < 0.01.

**Table 1 nanomaterials-13-00767-t001:** Phenotypic analysis of spleen, thymus, and bone marrow.

Organ	Parameter	Control (Mean ± SEM)	LD—TiO_2_ NPs (Mean ± SEM)	HD—TiO_2_ NPs (Mean ± SEM)
Spleen	CD3^+^	42.7 ± 2.0	36.5 ± 2.7 *	34.5 ± 3.4 *
	CD3^+^CD4^+^	26.4 ± 1.6	21.8 ± 2.2 *	22.5 ± 2.4
	CD3^+^CD8^+^	31.1 ± 2.3	26.6 ± 2.8	27.4 ± 3.0
	CD3^−^CD19^+^	34.7 ± 2.8	38.0 ± 2.7	36.6 ± 2.1
	CD3^−^CD335^+^	1.5 ± 0.2	2.7 ± 1.2	1.4 ± 0.1
Thymus	CD3^+^	39.5 ± 3.4	43.6 ± 2.9	35.9 ± 2.8
	CD3^+^CD4^+^	33.5 ± 3.0	36.8 ± 2.7	31.0 ± 2.2
	CD3^+^CD8^+^	31.0 ± 2.9	36.2 ± 3.3	26.1 ± 3.0
Bone marrow	CD3^+^	3.6 ± 0.4	3.4 ± 0.5	3.7 ± 0.6
	CD3^−^CD19^+^	2.6 ± 0.4	2.8 ± 0.4	2.4 ± 0.5
	CD3^−^CD335^+^	1.7 ± 0.3	1.2 ± 0.2	1.4 ± 0.3

Organs were derived from mice exposed to TiO_2_ NPs for seven weeks and controls. Cells were labeled with fluorescent monoclonal antibodies and analyzed by using flow cytometry. Control, control group (n = 10); LD–TiO_2_ NPs, group exposed to the low dose of TiO_2_ NPs (0.012 µg TiO_2_/g _b.w._) (n = 7–10); HD–TiO_2_ NPs, group exposed to the high dose of TiO_2_ NPs (0.958 µg TiO_2_/g _b.w._) (n = 10). CD3^+^, T-lymphocytes; CD3^+^CD4^+^, T-helper lymphocytes; CD3^+^CD8^+^, T-cytotoxic lymphocytes; CD3^−^CD19^+^, B-lymphocytes, CD3^−^CD335^+^, NK-cells. Results are expressed as the mean group percentage of labeled cells (mean ± SEM). Significance: * *p* < 0.05.

**Table 2 nanomaterials-13-00767-t002:** Concentrations of cytokines and chemokines in spleen cell culture supernatants in mice after seven-week inhalation of TiO_2_ NPs.

Cytokine	Control (Mean ± SEM)	LD—TiO_2_ NPs (Mean ± SEM)	HD—TiO_2_ NPs (Mean ± SEM)
IL-2	287.0 ± 102.0	276.1 ± 88.8	180.8 ± 54.7
IL-4	713.6 ± 189.1	965.2 ± 236.4	315.4 ± 121.5 *
IL-6	94.4 ± 33.4	52.9 ± 8.1	43.0 ± 12.0
IL-10	77.2 ± 28.0	54.1 ± 17.5	46.4 ± 21.5
IL-13	166.6 ± 74.2	97.8 ± 41.8	73.2 ± 40.8
IL-17A	63.8 ± 23.8	44.1 ± 26.9	30.7 ± 20.9
IL-18	1808.4 ± 301.8	2161.7 ± 264.4	1139.1 ± 226.6 *
IFN-γ	262.2 ± 159.0	134.1 ± 41.9	104.4 ± 54.7
TNF-α	56.0 ± 15.6	43.6 ± 6.1	31.8 ± 6.0
GM-CSF	25.0 ± 11.4	12.2 ± 3.1	9.8 ± 3.5
Eotaxin/CCL11	40.5 ± 11.2	34.7 ± 4.5	24.5 ± 6.0
MIP-1α/CCL3	127.2 ± 40.4	98.7 ± 12.0	77.0 ± 22.8
MIP-1β/CCL4	272.4 ± 103.9	200.9 ± 27.7	141.1 ± 34.9
MIP-2/CCL8	8.8 ± 2.4	5.4 ± 0.6	6.2 ± 0.9
RANTES/CCL5	68.8 ± 12.7	52.3 ± 6.8	43.4 ± 6.4
IP-10/CXCL10	122.1 ± 41.1	129.2 ± 44.6	88.1 ± 30.8
MCP-1/CCL2	216.5 ± 34.6	433.3 ± 110.6	243.6 ± 60.6
MCP-3/CCL7	71.4 ± 19.5	119.0 ± 40.0	82.6 ± 31.2

Cytokines were measured by using the luminescence method. Control, control group (n = 5–9); LD–TiO_2_ NPs, group exposed to the low dose of TiO_2_ NPs (0.012 µg TiO_2_/g _b.w._) (n = 4–7); HD–TiO_2_ NPs, group exposed to the high dose of TiO_2_ NPs (0.958 µg TiO_2_/g _b.w._) (n = 5–10). Results are expressed in pg/mL as mean group levels of cytokines (mean ± SEM). IL, interleukin; IFN, interferon; TNF, tumor necrosis factor; GMS-CSF, granulocyte-macrophage colony-stimulating factor; eotaxin, eosinophil chemotactic protein; MIP, macrophage inflammatory protein; RANTES, regulated on activation normal T-cell expressed and secreted; IP, interferon gamma-induced protein; MCP, monocyte chemotactic protein. Significance: * *p* < 0.05.

**Table 3 nanomaterials-13-00767-t003:** Antioxidant status (reduced glutathione and oxidized glutathione) in the blood of mice after seven-week inhalation of TiO_2_ NPs.

Parameter	Control (Mean ± SEM)	LD—TiO_2_ NPs (Mean ± SEM)	HD—TiO_2_ NPs (Mean ± SEM)
GSH	785.5 ± 110.4	1037.6 ± 165.9	1368.6 ± 108.5 **
GSSG	60.3 ± 3.5	61.5 ± 7.0	61.4 ± 5.1
GSH/GSSG	5.3 ± 0.7	7.6 ± 1.1 *	10.5 ± 1.1 **

Blood was derived from mice exposed to two different doses of TiO_2_ NPs and controls. Control, control group (n = 8); LD–TiO_2_ NPs, group exposed to the low dose of TiO_2_ NPs (0.012 µg TiO_2_/g _b.w._) (n = 8); HD–TiO_2_ NPs, group exposed to the high dose of TiO_2_ NPs (0.958 µg TiO_2_/g _b.w._) (n = 8). GSH, reduced glutathione; GSSG, oxidized glutathione. Results are expressed in μmol/L as mean values of GSH and GSSG (mean ± SEM). Significance: * *p* < 0.05, ** *p* < 0.01.

## Data Availability

The datasets generated and analyzed during the current study are available from the corresponding author on reasonable request.

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
