# Peer review of "Titanium Dioxide Nanoparticles Modulate Systemic Immune Response and Increase Levels of Reduced Glutathione in Mice after Seven-Week Inhalation"

_nanomaterials, 2023, doi:10.3390/nano13040767_

Round 1

Reviewer 1 Report (Previous Reviewer 1)

The authors simply refused to put in extra effort to improve its original submission. It is also questionable to divide data into different manuscript while they all falls in the same idea of understanding the toxicological impact of TiO2 after sub-chronic exposure.

Author Response

Reviewer 2 Report (New Reviewer)

"Titanium dioxide nanoparticles suppress the systemic immune 2 response and stimulate the antioxidant defense in mice after 3 seven-week inhalation"

This manuscript the use of TiO2-NP. In this study, the profile of lymphocyte markers, functional immune assays, and antioxidant defense markers were analyzed to assess the possible adverse effects of seven-week inhalation exposure to two different concentrations of TiO2 NPs (0.00167 and 0.1308 mg TiO2/m3 ) 41 in mice.

The Manuscript is good and worth presenting which will be good addition in the literature however, there are few queries which need to be addressed as;

1.      How the synthesis procedure differs from the previously reported methods? Significance Should be highlighted with the relevant literature.

2.      The author claimed that “, size was measured through TEM” The exact particle size should clearly be mentioned.

3.      UV/Vis results must be included in the main text of the manuscript.

4.      Recent/relevant references may be cited as;

https://link.springer.com/article/10.1007/s13204-021-02169-9

https://www.hindawi.com/journals/bca/2017/4101735/

https://link.springer.com/article/10.1007/s10904-018-0812-0

https://jnanobiotechnology.biomedcentral.com/articles/10.1186/s12951-018-0376-8

Author Response

Reviewer 3 Report (New Reviewer)

Titanium dioxide nanoparticles have many applications and multiple targets. From toxicological view point, they have represented a debated topic in the literature considering that they can cause oxidative stress, inflammation and apoptosis.

The title of the paper should indicate whether the reported effects were established at low doses or at high doses of titanium dioxide nanoparticles.

Can the text "stimulate the antioxidant defense" in the Title be exchanged by "increase the glutathione GSH levels"?

The Abstract (line 37) needs to indicate in which context the study was performed (e.g. diseases caused by TiO2 nanoparticles, danger of exposure to TiO2 nanoparticles, therapeutic approaches requiring inhalation of TiO2 nanoparticles,or other).

Was the suppression of the systemic immune response a desired effect (purpose of the study) or this is the resulting effect found after the TiO2 nanoparticle inhalation?

The terminology "antioxidant status of blood" is not clear to broad readers. The measured compounds should be indicated.

Considering the published impairments and lesions of the central nervous system after intranasal administration of TiO2 nanoparticles, the possible effects on the central nervous system should be mentioned.

Round 2

Reviewer 2 Report (New Reviewer)

Accepted in present form

Author Response

Dear Reviewer,

We would like to express our great appreciation to you for the comments on our manuscript. We hope that we have covered all the requirements and the manuscript is now acceptable for publishing.  

Answer:

The English language and style have been revised. All changes made in the text are visible in the “track changes” version of the manuscript. Our manuscript was revised by English native speaker prof. AR Collins acknowledged at the end of the manuscript.

This manuscript is a resubmission of an earlier submission. The following is a list of the peer review reports and author responses from that submission.

Round 1

Reviewer 1 Report

This manuscript by Mikusova et al presented their study on the systemic immune response exerted by low dose of TiO2 after seven-week inhalation. Given the extensive use of TiO2 in various industrial sectors, it is important to obtain detailed toxicological data on both acute and chronic exposure scenarios. This study certainly falls in this particular area. However, there are a few issues should be carefully addressed before it could be accepted for publication.

1. The data quality should be significantly improved. For example, the quality of the TEM images of TiO2 was poor. The size distribution diagram should be processed before presentation, rather than to use the raw data form of the instrument.

2. the authors should include more in vivo data to the manuscript, besides a few tables. What about the lung sections? What about the cytokine expressions? What about western blotting for the key proteins?

3. One main concern is the lack of proper controls. The authors should have positive and negative controls to compare with the effects of TiO2. Without them, it is impossible to come up with the conclusion presented in this study. 

Reviewer 2 Report

1. The antioxidant defense mechanism needs to be studied in detail. Some references can be made in this regard, for example: Phytotherapy Research, 2021, 35(6), pp. 2890-2901; International Journal of Biological Macromolecules, 2018, 111, pp. 780-786; Food Chemistry 2022, 388,133000.   2. Biological effects of nanoparticles should be discussed, see reference:Journal of Controlled Release, 2018, 278, pp. 122-126.   3. There is still not enough work to go around. The biggest advantage of nanoparticles can be used as carriers. An antioxidant can be selected to be immobilized on the nanoparticles to compare their biological effects.

Reviewer 3 Report

The manuscript reports the effects of sub-chronic inhalation exposure to two different concentrations of TiO2 NPs (0.00167 and 0.1308 mg TiO2/m3) carried out in whole-body inhalation chambers continuously for 7 weeks. Compared with previous studies on the systemic immune effects of TiO2 NPs using intratracheal or intranasal instillation, the whole-body inhalation experiment presented in this manuscript better imitates the physiological route of inhalation exposure to nanomaterials. This constitutes the significance  and novelty of this work and appropriateness for the special issue. The experiement is well designed and the results are well presented. However, although the conclusion states that "As the doses applied were comparable with concentrations to which humans may be exposed, our results may have serious implication for public health.", the discussion part of this manuscript just compares the results of this work to those of previous studies without proposing what the implication in public health is. In this regard, the reader could not really learn anything other than what are increased and what are decreased. It would be more signifciant if some useful messages could be gained from this work. The significance/impact of this manuscript could be raised if more discussion on this could be added.

Round 2

Reviewer 1 Report

The authors refused to address any of the comments raised. No improvements were provided.

Reviewer 2 Report

The application aspect is too simple and needs to be strengthened.